# Effects of Adipose-Derived Stem Cells and Their Conditioned Medium in a Human Ex Vivo Wound Model

**DOI:** 10.3390/cells11071198

**Published:** 2022-04-02

**Authors:** Xiao Guo, Christoph Schaudinn, Ulrike Blume-Peytavi, Annika Vogt, Fiorenza Rancan

**Affiliations:** 1Clinical Research Center for Hair and Skin Science, Department of Dermatology, Venerology and and Allergy, Charité–Universitaetsmedizin Berlin, Corporate Member of Freie Universität Berlin, Humboldt-Universität zu Berlin, and Berlin Institute of Health, 10117 Berlin, Germany; xiao.guo@charite.de (X.G.); ulrike.blume-peytavi@charite.de (U.B.-P.); annika.vogt@charite.de (A.V.); 2Advanced Light and Electron Microscopy, Zentrum für Biologische Gefahren und Spezielle Pathogene 4 (ZBS4), Robert Koch Institute, 13353 Berlin, Germany; schaudinnc@rki.de

**Keywords:** adipose-derived stem cells, conditioned medium, wound healing, Wnt/β-catenin, angiogenesis, ex vivo wound models

## Abstract

Adult stem cells have been extensively investigated for tissue repair therapies. Adipose-derived stem cells (ASCs) were shown to improve wound healing by promoting re-epithelialization and vascularization as well as modulating the inflammatory immune response. In this study, we used ex vivo human skin cultured in a six-well plate with trans-well inserts as a model for superficial wounds. Standardized wounds were created and treated with allogeneic ASCs, ASCs conditioned medium (ASC-CM), or cell culture medium (DMEM) supplemented with fetal calf serum (FCS). Skin viability (XTT test), histology (hematoxylin and eosin, H and E), β-catenin expression as well as inflammatory mediators and growth factors were monitored over 12 days of skin culture. We observed only a moderate time-dependent decrease in skin metabolic activity while skin morphology was preserved, and re-epithelialization occurred at the wound edges. An increase in β-catenin expression was observed in the newly formed epithelia, especially in the samples treated with ASC-CM. In general, increased growth factors and inflammatory mediators, e.g., hepatocytes growth factor (HGF), platelet-derived growth factor subunit AA (PDGF-AA), IL-1α, IL-7, TNF-α, and IL-10, were observed over the incubation time. Interestingly, different expression profiles were observed for the different treatments. Samples treated with ASC-CM significantly increased the levels of inflammatory cytokines and PDGF-AA with respect to control, whereas the treatment with ASCs in DMEM with 10% FCS resulted in significantly increased levels of fibroblast growth factor-basic (FGF-basic) and moderate increases of immunomodulatory cytokines. These results confirm that the wound microenvironment can influence the type of mediators secreted by ASCs and the mode as to how they improve the wound healing process. Comparative investigations with pre-activated ASCs will elucidate further aspects of the wound healing mechanism and improve the protocols of ACS application.

## 1. Introduction

Wound healing is a complex and dynamic physiological process requiring different types of skin and blood cell populations involved in four sequential but overlapping phases: hemostasis, inflammation, proliferation, and remodeling [1]. Impairments in the wound healing process can lead to hypertrophic scars or chronic non-healing wounds [2,3,4]. Chronic wounds are defined as wounds with a delayed healing process (more than 12 weeks). In such lesions, the transition from the inflammatory to the proliferative phase is compromised [5]. This leads to a dysregulation in the proliferation and migration of skin cells at the injured site resulting in insufficient extracellular matrix (ECM) deposition and dysfunctional re-epithelialization [6]. The etiology of chronic wounds is multifactorial and is related to hyperinflammatory processes, oxidative stress, local tissue hypoxia, immunosuppression (age, systemic illnesses), neuropathy (diabetes), wound colonization or infection [7,8]. Consequently, most of the current therapies are focused on improving blood supply, preventing or treating infections, regulating the inflammatory and oxidative processes, and finally promoting angiogenesis and re-epithelialization [9].

An increasing number of stem cell-based therapies have been developed for tissue engineering and regenerative medicine [10]. Especially mesenchymal stem cells (MSCs) have been extensively investigated in this field. They can be isolated from different body compartments, including bone marrow, adipose and muscle tissue, or peripheral blood, and can differentiate into adipocytes, chondrocytes, and myocytes [11]. In addition, they have the ability to boost tissue regeneration and to modulate inflammatory immune responses in a paracrine mode thanks to released trophic and immune active factors [12,13,14]. Particular interest has been given to adipose-derived stem cells (ASCs). Besides the properties common to other MSCs, such as self-renewal, multilineage differentiation, and low immunogenicity [15], ASCs are promising candidates for clinical applications due to their availability and abundancy [6]. The use of ASCs in wound healing therapies is supported by their anti-inflammatory properties along with their capacity to enhance the migration and proliferation of vascular endothelial cells, fibroblasts, and epithelial cells [6,16,17]. ASCs have been also shown to regulate keratinocytes’ proliferation and other wound healing processes by modulating the expression of β-catenin, a crucial mediator in the canonical Wnt signaling pathway [18]. The mechanism by which ASCs orchestrate tissue repair and regeneration is mainly accomplished by secretion of growth factors, such as fibroblast growth factor (FGF), hepatocyte growth factor (HGF), and platelet-derived growth factor (PDGF) [19]. In addition, ASCs can release immunomodulatory cytokines that can restrain or fortify the immune response depending on the specific inflammatory microenvironment [20,21,22]. Finally, they were shown to participate in tissue repair by differentiating into different cell types upon direct cell-to-cell contact [23,24].

However, some drawbacks limit the clinical application of stem cells. On the one hand, there are issues concerning decreased cell viability and proliferation after complex preparation and storage procedures. On the other hand, there are still unsolved safety-related complications such as genetic instability and undesirable malignant differentiation [22]. For these reasons, ASCs’ extracellular vesicles (EVs) and conditioned medium from ASCs’ culture (ASC-CM) have gained considerable interest as cell-free treatments [25]. ASC-CM contains healing promoting proteins (e.g., cytokines, growth factors), nucleic acids (non-coding RNA), and lipids (e.g., prostaglandins) that are present as soluble factors or loaded into EVs [26,27].

Several in vitro and in vivo animal studies have shown the beneficial effects of ASCs and ASC-CM on wound healing [25,28,29]. However, it is known that skin morphology, hair follicle density, immune response, and some aspects of the wound healing process in rodents are different from those in humans. There are very few studies using human derived models. For example, only one study showed the stimulating effects of ASCs in the wound healing process using a human ex vivo model for burn wounds [30]. In this study, we used an ex vivo superficial wound model that was previously established in our group [31,32] to better understand the biological properties of ASCs and ASC-CM in human wound healing. Besides monitoring the re-epithelialization process, we also investigated the expression of β-catenin and the levels of growth factors and immunomodulatory cytokines. To our knowledge, this is the first study where a human ex vivo model was used to investigate the beneficial effects of ASCs and ASC-CM by monitoring the main biological mediators involved in wound healing.

## 2. Materials and Methods

### 2.1. Isolation and Culture of ASCs

ASCs were isolated from the abdominal fat tissue of healthy female donors, who underwent cosmetic surgery. The experimental protocol was written following the guidelines of the Declaration of Helsinki and was approved by the Ethics Committee of the Charité–Universitätsmedizin Berlin (approval EA1/382/20, February 2021). Adipose tissue was accurately minced with scissors and washed with Dulbecco’s phosphate-buffered saline (PBS; Biochrom, Berlin, Germany), containing 1% penicillin-streptomycin (Sigma-Aldrich, Hamburg, Germany). The tissue (30 mL) was digested with 0.2% collagenase I (0.2% collagenase I: adipose tissue = 1:3) (Worthington Biochemical Corporation, Lakewood, NJ, USA) at 37 °C for 30–45 min. After digestion, samples were centrifuged at 300× *g* for 10 min and the upper undigested tissue was discharged. The cell pellet was resuspended in PBS and filtered through a 70 μm filter (Falcon^TM^, Durham, NC, USA) to remove residues of connective tissue. After another centrifugation step, approximately 2 × 10^6^ cells were obtained and resuspended in Dulbecco’s modified medium (DMEM; Lonza^TM^, Verviers, Belgium) with 4.5 g/L glucose and 2 mM UltraGlutamine I and supplemented with 10% fetal calf serum (FCS; PAA, Heidelberg, Germany), 100 I.E./mL penicillin, and 100 g/mL streptomycin (Sigma-Aldrich, Hamburg, Germany). The cells were seeded in T75 flasks and cultured in an incubator at 37 °C, 95% relative humidity, and 5% CO_2_. After 48 h incubation, non-adherent cells were discarded and fresh culture medium was added. Cell subculture was performed when ASCs reached approximately 80% confluence. ASCs from 4 different donors (female, aged 34–42) of passages 3 to 7 were used for the experiments.

### 2.2. Characterization of ASCs

Immunofluorescence staining was performed to reveal the morphology of living cells. ASCs (1 × 10^4^ cells/mL) at passage 4 were seeded in a µ-Dish, 35 mm in diameter (Ibidi GmbH, Gräfelfing, Germany) and incubated at 37 °C, 5% CO_2_, and 95% relative humidity for 48 h. After a washing step with DPBS, Concanavalin A (Con A) conjugated to AlexaFluor 488 (Thermo Fisher Scientific, Waltham, MA, USA; 100 μg/mL) was used to stain the plasma membrane of ASCs (incubation at room temperature for 20 min). Following two washing steps in DPBS, the perinuclear membranes were then stained with SYTO 62 (Thermo Fisher Scientific, Waltham, MA, USA; 5 µM) for 10 min at room temperature. After washing two times in dH_2_O, stained cells were supplied with 2 mL culture medium and covered with a polymer coverslip. Images were captured by means of a confocal laser microscope (LSM Exciter, Jena, Germany).

Immunostaining and flow cytometry were used to identify the phenotype of ASCs and show the presence of surface markers typical of mesenchymal stem cells like CD29, CD44, and CD90 as well as the absence of CD45. ASCs were trypsinized, centrifuged, re-suspended in 3% bovine serum albumin (BSA; Biomol, Hamburg, Germany) to get a 0.5–1 × 10^6^ cell/mL suspension, and incubated at room temperature for 10 min. After centrifugation, ASCs were resuspended in 200 µL of 3% BSA and stained with 5 µL APC-labeled anti-CD44 (Biolegend, San Diego, CA, USA), 5 µL PE-labeled anti-CD90 antibodies (Biolegend, San Diego, CA, USA), and 5 µL Alexa 488-labeled anti-CD29 (Biolegend, San Diego, CA, USA) or 5 µL of fluorescein isothiocyanate (FITC)-labeled anti-CD45 (Biozol, München, Germany) antibodies. Cells were incubated in a Thermo-Mixer at 4 °C and 700 rpm for 20 min. Excess antibodies were washed with cold PBS. ASCs were fixed with 4% PFA for 5 min, washed, and resuspended in PBS. The fluorescence intensity of ASCs was measured using a flow cytometer (FACS Calibur, Becton Dickinson, Heidelberg, Germany) and analyzed with the FCS Express software version 3.1 (De Novo Software, Glendale, CA, USA). At least 20,000 events were collected. Unstained cells were used as negative reference to define gating and background fluorescence. Cells were also imaged using a confocal laser scan microscope LSM 700 (Carl Zeiss Microscopy GmbH, Jena, Germany) coupled to a CCD camera.

### 2.3. Preparation of Conditioned Medium

ASCs at 80% confluence were washed with PBS without calcium and magnesium and cultured in FCS-free DMEM for a further 48 h at 37 °C in 95% relative humidity and 5% CO_2_. ASC-CM was collected, centrifuged at 300× *g* for 5 min, and filtered through 0.22 μm pore size filters (Carl Roth GmbH, Karlsruhe, Germany). ASC-CM was aliquoted and stored at −80 °C for further use.

### 2.4. Ex Vivo Wound Models

Ex vivo wound models were created using abdominal skin, obtained from female donors that had undergone abdominoplasty with an average age of 38 (29–47 years old). The experimental protocol was written following the guidelines of the Declaration of Helsinki and was approved by the Ethics Committee of the Charité–Universitätsmedizin Berlin (approval EA1/382/20, February 2021). Written informed consent were signed by all donors. Skin samples were processed within 2–4 h after surgery and examined to select intact skin without stretch marks and scars. Samples were then cleaned with sterile 0.9% saline solution and cut into 1.5 × 1.5 cm pieces. Subcutaneous adipose tissue was almost completely removed, except for a 5 mm layer. Each skin sample was stretched on a styrofoam block covered with Parafilm (Bemis Company, Neenah, WI, USA) and fixed with syringe needles. A superficial wound with a diameter of 5 mm was created by removing the epidermis with a ball-shaped milling cutter (No. 28725, Proxxon, Föhren, Germany), rotating at 16,000 rpm and mounted on a micro motor handpiece (Marathon N7, TPC Advanced Technology, Inc., Diamond Bar, CA, USA). The skin samples were then transferred on inserts having a membrane with 8 µm pore size (BD Falcon™, Durham, NC, USA). The inserts were positioned in six-well culture plates (BD Falcon™) where each well was filled with 2 mL of DMEM medium supplemented with 2 mM UltraGlutamine I, 4.5 g/mL glucose, 10% FCS, 100 I.E./mL penicillin, and 100 g/mL streptomycin.

### 2.5. Topical Treatments of Ex Vivo Wounds

In each experiment, 20 skin tissue pieces were divided into four groups (one for control and three for treatments) with 5 pieces each that were cultured for different times: 1, 3, 6, 9, and 12 days (Figure 1). At day 0, wounds were treated with 20 µL of: (i) ASCs (1 × 10^6^ cells/mL); (ii) ASC-CM; or (iii) DMEM supplemented with 10% FCS. The group with untreated wounds served as control. Every day, starting from day 1, 10 µL of DMEM with 10% FCS was added to the ASCs and DMEM groups, whereas 10 µL of conditioned medium was added to the ASC-CM group. One milliliter of supplemented DMEM culture medium was replaced every day. All groups were incubated at 37 °C in 5% CO_2_ and 95% relative humidity. At each time point (day 1, day 3, day 6, day 9, and day 12), one skin sample and one milliliter of culture medium per group was harvested for further investigations and stored or processed as illustrated in Figure 1.

### 2.6. XTT Assay

At each time point, skin explants were collected, placed in a six-well plate, and incubated in 500 µL of 2,3-bis-(2-methoxy-4-nitro-5-sulfophenyl)-2H-tetrazolium-5-carboxanilide (XTT, Roche Diagnostic, Berlin, Germany), for 2 h at 37 °C in 5% CO_2_ and 95% relative humidity. Thereafter, 2 × 100 µL of XTT were collected from each sample, placed in a 96-well microplate in duplicate, and the absorbance was measured using a 2300 EnSpire microplate reader (PerkinElmer, Santa Clara, CA, USA). Sample optical density (OD) was measured at wavelengths of 450 nm, using 650 nm as the reference wavelength. All values were normalized to the untreated group at day 1. Results of the metabolic activity of skin cells are presented as dot plots. Six independent experiments are reported along with the median and range.

### 2.7. Preparation of Cryosections and Skin Extracts

After the XTT assay, each skin sample was cut into two halves (Figure 1). One piece was immersed in tissue freezing medium (HM560 Cryo-Star Microm Laborgeräte GmbH, Walldorf, Germany), plunge-frozen in liquid nitrogen, and further processed for histology. Skin samples were cut vertically in 7 µm thick slices using a microtome (Frigocut 1510 S, Leica, Bensheim, Germany). At least ten wound sections from the wound center and edges were prepared for each skin sample and used for H and E or β-catenin staining and further microscopic analysis. The other half of the skin sample was used to prepare protein extracts. It was transferred to a 2 mL Eppendorf tube, frozen in liquid nitrogen, and kept at −80 °C until it was processed as follows: skin sections were cut horizontally using a microtome (20 × 50 µm corresponding roughly to the total layer of epidermis and most of the dermis). Sections were put in 500 µL extraction buffer (100 mM Tris-HCl; 150 mM NaCl; 1 mM EDTA; 1 g Triton-X-100) and incubated for 1.5 h in a Thermo mixer at 4 °C and 700 rpm. Samples were then sonicated at 4 °C for 10 min (37 Hz, 200 Weff), vortexed, and centrifuged at 450× *g* for 5 min. The supernatant was aliquoted and stored at −80 °C for further measurements.

### 2.8. Histology and Immunofluorescence Microscopy

The histological analysis of skin samples was performed after hematoxylin and eosin (H and E) staining. All sections were fixed with acetone for 10 min at room temperature and were incubated with the two dyes following the manufacturer’s instructions (Roth, Karlsruhe, Germany). Images were captured using an Olympus IX 50 microscope (OLYMPUS, Hamburg, Germany).

Immunofluorescent staining was conducted for the detection of β-catenin. Acetone-fixed tissue sections were blocked with a serum-free protein blocking kit (DaKo, Glostru, Denmark) for 1 h. Sections were then incubated overnight using anti-β-catenin polyclonal antibody (Thermo Fisher Scientific, Waltham, MA, USA; 1:200 in 5% FCS-PBS). After washing, slides were incubated with FITC-conjugated goat anti-rabbit IgG (Vector Laboratories, Burlingame, CA, USA; 1:50 in 5% FCS-PBS) for 45 min, followed by three washing steps in dH_2_O. Cell nuclei were stained using propidium iodide (PI; 1:100 in dH_2_O) for 20 min at room temperature. After washing, mounting medium (Vector Laboratories, Burlingame, CA, USA) was dropped onto the sections and coverslips were applied. Images were captured by means of a confocal laser microscope (LSM Exciter, Jena, Germany). For each sample and time point, 15 to 35 randomly chosen visual fields from different sections and donors were taken. The mean fluorescence intensity (MFI) of β-catenin staining in the viable epidermis and wound edges was quantified using the ImageJ software, version 1.47 (National Institute of Health, Bethesda, MD, USA).

### 2.9. Multiplex Assay

Twelve selected cytokines and growth factors (VEGF, PDGF-AA, PDGF-BB, HGF, FGF-basic, EGF, TGF-β1, IL-1α, IL-1β, IL-7, IL-10, and TNF-α) were measured in skin extracts and culture medium using a LEGENDPlex custom made kit (Biolegend, San Diego, CA, USA). In this bead-based multiplex immunoassay, each bead is marked by a defined amount of allophycocyanin (APC) and conjugated with a capture antibody specific for a target protein. After incubation with skin extracts or medium, the addition of detection antibodies and streptavidin-phycoerythrin (SA-PE) generates a unique fluorescence signal for each target protein, which is in proportion to the antigen concentration. Each sample and standard were run in duplicate. The fluorescence intensity was detected by flow cytometry (FACS Calibur, BD, Germany). At least 400 events in the gated region were measured. The concentration of each analyte was calculated using LEGENDplex software (https://www.biolegend.com/legendplex/software, last access on 27 December 2021). The analyte concentration in skin extracts was normalized to the total protein content, which was assessed by the Pierce 660 nm Protein Assay (Thermo Fisher Scientific, Waltham, MA, USA). Absorbance values were determined using a plate reader (EnSpire Multimode Perkin Elmer, Akron, OH, USA). The data are shown as increments of each analyte over time.

### 2.10. Statistical Analysis

Skin samples from a total of 16 donors were used in this study. Skin from four donors was used for the measurements of β-catenin in wound extracts (see Appendix A). Skin from other six donors was used in preliminary experiments to set up the experimental protocol and measure IL-6 and IL-8 (see Appendix A). Finally, skin from other six donors was used in six independent experiments for histology, viability, immunostaining as well as cytokines and growth factor analyses. For the evaluation of β-catenin expression in wound sections, at least 15 images were analyzed for control (D9 and D12), 20 images for ASCs (D3, D6, D9), 25 images for DMEM (D9, D12) and control (D1, D3, D6), and 30 images for ASC-CM at all time points. Data were presented as dot plots, diagrams, or violin plots and were prepared using GraphPad Prism (GraphPad Software, San Diego, CA, USA). The statistical analysis was performed with SPSS 23.0 (IBM SPSS Statistics 23.0) using the Wilcoxon signed-rank test for non-normally distributed paired samples, and the Mann–Whitney *U* test for unpaired nonparametric values to illustrate differences between the two groups. Statistical significance was represented as follows: * *p* < 0.05, ** *p* < 0.01.

## 3. Results

### 3.1. Characterization of ASCs

Prior to the preparation of conditioned medium and the use of ASCs and ASC-CM in the ex vivo wound experiments, cell morphology and their predominant phenotype at passage four and three were identified (Figure 2). The plasma membrane of adherent ASCs was evidenced by staining with concanavalin A (green) and the perinuclear membranes were stained in red with Syto62 (Figure 2A). The adherent-growing ASC population exhibited the expected spindle-like morphology typical of mesenchymal stem cells. After immunostaining, both microscopy (Figure 2B) and flow cytometry (Figure 2C) confirmed that the ASCs expressed typical mesenchymal stem cells surface markers like CD29, CD44, and CD90. On the contrary, CD45, one of the major surface markers for hematopoietic stem cells, was not expressed.

### 3.2. Ex Vivo Skin

To evaluate the possible beneficial effects of ASCs and ASC-CM on cutaneous wound healing, an ex vivo wound model, based on full-thickness human skin and maintained in supplemented culture medium, was established. After preliminary experiments for optimizing the organ-culture conditions, the treatment protocol was chosen in which treatments, as well as medium replacement, were performed every day, whereas sample collection was done every 3 days (Figure 1). The daily replacement of 1 mL medium allowed for an optimal level of nutrients. The collection and analysis of both medium and skin samples every three days allowed us to follow the temporal evolution of multiple parameters and correlate them to the wound healing process.

### 3.3. Identification of ASCs in Wound Sections

To trace the ASCs after their topical application and identify them in the wound tissue, we stained the sections with antibodies specific for CD44 and CD29 (Figure 3). ASCs positive for both CD44 and CD29 were visible on the top of the wound at all incubation times. These images indicated that most of ASCs remained on the top of the wound without differentiating into skin cells. Interestingly, in some sections we found an increased amount of CD44-negative and CD29-positive cells at the wound edges (e.g. Figure 3, day 9). These cells might be keratinocyte stem cells that were activated by ASCs and had migrated to the wound edge. Alternatively, they might be ASCs that have migrated into the wound tissue and downregulated the CD44 glycoprotein. Thus, further experiments are still necessary to understand the fate of ASCs after topical application to wounds.

### 3.4. Histological Analysis

Skin morphology and the re-epithelialization processes were monitored in H and E stained cryosections. Representative pictures are shown in Figure 4A. The morphology of the epidermal layer in regions close to the wound edges changed during the incubation time. The thickness of the stratum corneum and stratum granulosum increased over time, indicating that the proliferation at the basal cell layer along with the differentiation processes in the upper layers continued during the 12 culture days. Starting from day 9, in DMEM and untreated samples, severe disruption of the stratum granulosum and stratum corneum occurred, which were probably accentuated through the cryosectioning procedure. This reduced stability of the tissue samples was indicative of a loss of intercellular adhesion. On the contrary, a better tissue integrity could be observed in ASCs and ASC-CM samples even after 12 days of culture.

After 9-days of incubation, the formation of a new epithelium starting from the wound edge was visible. A newly formed epithelium of two to four cell layers was visible in most of the samples at day 12. Partial re-epithelialization in the wound center could be seen especially in ASCs-treated samples (Figure 4B). ASCs were always observed on the top of the wound over the entire time course of the experiment (Figure 4C), as also shown in the wound sections immunostained with anti-CD44 and anti-CD29 antibodies (Figure 3). In summary, these results showed that the re-epithelialization process took place in all ex vivo samples. Nevertheless, the application of ASCs or ASC-CM resulted in a superior maintenance of tissue morphology and integrity over the 12-days culture at the air-liquid interface.

### 3.5. Reductive Activity of Ex Vivo Skin

The reductive activity in ex vivo skin was measured every 3 days using the XTT assay. As the dot plot in Figure 4D shows, after 3 days of incubation a clear decrease in skin activity was detected with respect to that on day 1, following a more moderate decrease on day 6, until approximately 50% of the initial value was reached on day 9 and 12.

No significant differences were found between each group at every time point, with the only exception the ASCs group with respect to the control group at day 9 (*p* = 0.028). A strong variability was found between the donors, especially on day 1, 3, and 6, whereas the values were less scattered on day 9 and 12, indicating that a sort of steady state had been reached.

### 3.6. Epidermal β-Catenin Expression in Ex Vivo Skin

There is increasing evidence that β-catenin plays a central role in mediating skin regeneration, e.g., by promoting the proliferation of stem cells of hair follicles and of the basal layer in the inter-follicular epidermis [33]. These cells may be involved in the formation of new hair follicles during tissue regeneration without the formation of scar tissue [34].

To evaluate the effect of the topical treatments on the canonical Wnt signaling pathway, β-catenin expression in the cultured ex vivo wounds was measured by ELISA. Tissue extracts from four donors were analyzed at all time-points (see Appendix A). β-catenin expression levels remained constant until day 6, slightly decreased at day 9 and reached 50% of the initial levels at day 12. Interestingly, the β-catenin curve time course was similar to that of the XTT reduction assay (Figure 4D). In the whole tissue extracts, no significant differences were measured between the treatment and the control groups. Next, we tested whether local increments of β-catenin, e.g., at the wound edges, had occurred using specific immunofluorescence staining of wound sections. Figure 5A showed representative pictures of epidermis in areas close to the wound edge at all incubation times, whereas Figure 5B showed representative pictures of newly formed epithelia at the edge and in the center of the wounds at day 12. In the tissue close to the wound edges (Figure 5A), the β-catenin-related signal was mainly located in the viable epidermis layers, whereas only a weak signal was detected in the stratum granulosum and stratum corneum, where cells are in an advanced differentiation stage. Low signal was measured in the dermis and in the cell nuclei. For all treatment and control groups, a slight decrease in the β-catenin signal was observed starting from day 3. In ASC-CM treated skin, a relatively higher β-catenin signal in comparison to the other groups was observed at each time point. Interestingly, in ASCs-treated samples, a few layers of β-catenin-positive stem cells could be seen on the top of the epidermis on day 1 and 3 (Figure 5A, arrows). Concerning the newly formed epithelial layers at day 12 (Figure 5B), the strongest β-catenin signaling was seen in ASCs and ASC-CM treated samples both at the edges and in the center of the wounds. Interestingly, the cell nuclei of the new epithelium had a strong PI staining in the DMEM, ASC-CM, and ASCs groups, whereas low PI signal was measured in the control group. To verify the existence of statistical differences between the treated and untreated wounds, the β-catenin-related MFI was measured in areas close to the wound edges using the ImageJ software (Figure 5C). Higher β-catenin-related MFI with respect to all other groups was detected in the ASC-CM group, especially during the wound healing phase (day 6 and 9). Significantly increased expression with respect to control was measured on day 6. For the ASCs group, although no significant differences were found, we observed relatively higher MFI with respect to untreated wounds, especially towards the end of the incubation time (Day 9, *p* = 0.053). These observations suggest that the canonical Wnt/β-catenin pathway is involved in the wound healing process and was more activated after the topical application of ASC-CM and ASCs.

### 3.7. Detection of Growth Factors in Ex Vivo Skin Extracts and Culture Medium

To elucidate the beneficial effects of ASCs and ASC-CM on the wound healing process in the ex vivo human skin, growth factors were analyzed using a multiplex assay in both skin extracts (Figure 6) and culture medium (Figure 7). Growth factors stimulate proliferation and differentiation of keratinocytes, fibroblasts, and endothelial cells contributing to vascularization and re-epithelialization during tissue repair [35]. Our results revealed the expression of VEGF, PDGF-AA, FGF-basic, and HGF in ex vivo skin. On the contrary, no PDGF-BB, EGF, and activated TGF-β1 could be detected, probably due to the absence of cells from the blood compartment. In skin extracts, high levels of VEGF and FGF-basic were measured, whereas PDGF-AA and HGF concentrations were much lower and TGF-β1 was detected only in two of six donors. In culture medium, only high values of VEGF were measured, with lower levels of PDGF-AA and HGF, and no detectable levels of FGF-basic and TGF-β1.

As shown in Figure 6D, a significantly higher expression of VEGF was observed in skin extracts following the daily topical treatment with DMEM with 10% FCS, especially on day 9 and 12. Also in the medium (Figure 7C), the DMEM group had the highest levels of VEGF as compared to the other groups. The ASCs and ASC-CM treatments had little additional effects with respect to control with regard to the expression of VEGF. Intriguingly, as showed in skin extracts (Figure 6D), at day 9 and 12, wounds treated with ASCs followed by daily application of 10 µL DMEM with 10% FCS had significantly lower levels of VEGF than the group treated with DMEM with 10% FCS only. In addition, at day 6, the ASCs group had the lowest values compared to all the other groups.

Significant increments of PDGF-AA with respect to controls were observed starting from day 9 in the extracts and day 6 in the culture medium (Figure 6C and Figure 7B). However, for this growth factor, the highest values were observed in the ASC-CM and ASCs treated groups. Especially in the ASC-CM group, significant increments with respect to control were observed in extracts at day 9 and 12 (Figure 6C) as well as in culture medium at day 6, 9, and 12 (Figure 7B). No significant increase in this growth factor with respect to the untreated control could be measured for the DMEM group, except at day 9 in skin extracts.

As for HGF, a remarkable increment was measured in skin extracts at an early time point. Its expression was significantly increased with respect to control only for the ASC-CM and ASCs treated groups but not in the DMEM group. In particular, in skin extracts higher values were measured after treatment with ASCs at day 1 and 6 (Figure 6B), while in the culture medium HGF had considerably higher concentrations in both the ASC-CM and ASCs treated groups at day 6, 9, and 12 (Figure 7A). The DMEM group showed no significant increases with respect to control.

Concerning FGF-basic, while high concentrations were measured in skin extracts, no signal could be detected in the culture medium. Interestingly, FGF-basic was significantly upregulated in the ASCs group compared to both the control and the other treatment groups (Figure 6A). On the contrary, no significant increases were measured for the DMEM and the ASC-CM groups. This finding indicates that ASCs produced FGF-basic or induced its production after their topical application and culture at the wound surface.

### 3.8. Detection of Inflammatory Cytokines in Ex Vivo Skin Extracts and Culture Medium

Besides growth factors, immunomodulatory cytokines were also measured using the multiplex assay (IL-1α, IL-1β, IL-7, IL-10, and TNF-α, Figure 8 and Figure 9). IL-6 and IL-8 were measured in preliminary experiments using a slightly different protocol (see Appendix A).

All cytokines could be measured in both tissue extracts and culture medium, with the exception of IL-1β, which was measured only in skin extracts, and IL-10, which was detected in culture media only. The highest cytokine amounts were measured for IL-6, IL-8, and IL-1α, followed by TNF-α and IL-10 (only in culture media), whereas very low, but constantly increasing levels of IL-1 β and IL-7 were observed.

In general, for the pro-inflammatory cytokine IL-1β a slight increase was observed following the treatments as compared to untreated control. (Figure 8A). However, significant increases were measured only for the DMEM group at day 3 and 12 as well as for the ASC-CM group at day 9. No IL-1β was detected in the culture media. As for TNF-α, increased values with respect to the other groups were detected for the ASC-CM treatment in the skin extracts (Figure 8B) as well as in the culture medium (Figure 9A). In tissue extracts, statistical significance was found only at day 6 and 12 as compared to ASCs and untreated groups, respectively. In medium samples, significantly higher values were calculated for the ASC-CM group with respect to the other treatment groups at all incubation times, whereas a significant increase with respect to control was found between day 6 and 12. No significant increase or decrease with respect to controls was evidenced for the ASCs and DMEM groups. As shown in Figure 8C, the expression levels of IL-1α in skin extracts were relatively low in almost all samples with the exception of a few donors with relatively high values. Small increases with respect to untreated samples were seen in each treatment group starting from day 9 with significantly increased values at day 12. As for the release in medium (Figure 9C), significantly higher levels with respect to control were observed for the ASC-CM and the ASCs groups already at day 6 and in the following days. Despite it being expressed at very low levels in all groups, IL-7 increased constantly over time. In skin extracts (Figure 8D), the secretion of IL-7 was significantly promoted in the ASCs treated group at day 6 and 9 and in the ASC-CM group at day 3 and 12. In the culture medium (Figure 9D), higher levels of IL-7 with respect to control were measured only in ASC-CM treatment group in the final incubation phase, i.e., at day 9 and 12. In general, no significant increases were measured for the DMEM-treated group with respect to control, indicating that the increments of IL-7 were due to the presence of ASCs or their conditioned medium. IL-10, was detected only in the culture medium and increased values were found for all treatment groups in comparison to untreated control (Figure 9B). In particular, the DMEM treatment group had significantly higher levels than the controls from day 3 to day 12. For the ASC-CM group, IL-10 was significantly increased with respect to the controls at day 3 and 9, whereas for the ASCs samples, significant increments were found at day 9 with respect to control and at day 12 with respect to the DMEM group. Finally, IL-6 and IL-8 were measured by ELISA in preliminary experiments where the wound had been treated with ASCs using a slightly different protocol (see Appendix A). In general, both IL-6 and IL-8 increased with time in both control and sample groups, but significant increases were found mainly for IL-8 in the epidermis of the ASCs-treated group with respect to the DMEM group (see Appendix A).

## 4. Discussion

In the last few years, several publications have reported that ASCs can promote tissue angiogenesis and re-epithelialization by secreting growth factors and immunomodulatory cytokines and even by differentiating into multiple cell types including keratinocytes and fibroblasts [6]. Nevertheless, despite very promising clinical trials, most of the in vivo studies exploring the mechanisms of action of ASCs in wound healing were conducted on in vivo murine models. Because the wound healing process in humans differs from that in mice [36], in our study we used a human wound model based on ex vivo skin, to test allogenic ASCs isolated from subcutaneous tissue. The stromal vascular fraction (SVF), which contains blood and endothelial cells as well as mesenchymal stem cells and fibroblasts [37], was isolated through enzymatic digestion. The purified ASCs population was obtained from the SVF by centrifugation and adherent cell culture. We chose the third to seventh generation of ASCs for subsequent experiments to minimize interference from other cell types and achieve a homogeneous cell population, as shown by the phenotype analysis (Figure 2). The ASCs and ASC-CM were applied topically on ex vivo wounds created on skin samples that were cultured at the air–liquid interface in six-well plates with trans-well inserts for a maximum of 12 days. Similar ex vivo wound models have been developed in the last years and were successfully used in a number of wound healing studies [38].

### 4.1. Effects on Skin Morphology and Re-Epithelialization

We first examined skin sections to evaluate skin morphology and the re-epithelialization process. In H and E-stained sections, the physiological morphology of epidermis and dermis was retained in the first 6 days. We only noticed morphological changes after 9 days of skin culture, e.g., loss of intercellular adhesion and disruption of the upper epidermal layers. Nevertheless, these effects were reduced in samples treated with ASCs and ASC-CM, indicating that the treatments positively influenced the maintenance of tissue integrity. In addition, we could observe the positive effects of ASCs and ASC-CM on the re-epithelialization process from the wound edges inwards (Figure 4A,B).

The XTT assay was used to measure the reductive activity of ex vivo skin [39]. The results show that skin exhibited a trend of time-dependent decrease in reductive activity until a steady state was reached between 9 and 12 days. Skin reductive activity is correlated to its overall oxidation state (e.g., levels of NADH or glutathione), cell metabolic activity, and the overall cell viability. The redox state of skin is closely linked to inflammatory processes. Namely, it has been shown that inflammatory processes ongoing during wound healing are associated with the production of radical species and the onset of an oxidative environment [40]. The latter might cause a decline of skin reductive activity. In addition, a decrease in cell metabolic activity may occur due to lack of essential nutrients for certain cell populations. Nutrient-sensing regulators respond to this depletion causing a down-regulation of the metabolic processes [41]. On the other hand, re-epithelialization, as well as expression of growth factors and cytokines, were ongoing over the 12 days of incubation, indicating that most of skin cell populations were metabolically active. Thus, the observed decrease in skin reductive activity is probably the result of different ongoing processes like oxidative/inflammatory processes along with reduced metabolic activity as well as apoptosis or necrosis.

### 4.2. Effects on β-Catenin

To further monitor the effects of the ASC-CM and ASCs treatments on the wound healing process, we measured the expression of β-catenin in wound tissue. We observed a time-dependent reduction in the β-catenin signal in the whole wound tissue (see Appendix A) and in the epidermis surrounding the wound (Figure 5A,C). This decrease correlated with that of skin reductive activity (Figure 4D) and confirmed a partial loss of cell metabolic activity and viability. Nevertheless, when comparing the different treatment groups, a relatively higher β-catenin expression with respect to control skin was measured in the ASCs and especially in the ASC-CM group. These results correlated with the histological observations of a better re-epithelialization and tissue integrity in the two treated groups.

β-catenin has two main functions: it is part of intercellular adherence junctions, and it acts as a crucial mediator in the canonical Wnt signaling pathway. In general, the Wnt/β-catenin pathway has been found to regulate wound repair by promoting the migration, adhesion, and retention of macrophages and fibroblasts in the granulation tissue [33,42,43]. Moreover, it was demonstrated that β-catenin could also coordinate the secretion of mediators such as TGF-β in the early wound healing phase and activate matrix metallo-proteinases (MMPs) in the later stages, thus remodeling the extracellular matrix [44]. More complex is the role of β-catenin in the epidermis of healing wounds. Controversial results have been published regarding the effects of β-catenin upregulation in keratinocytes. Some studies showed that increased nuclear levels of β-catenin induced keratinocyte proliferation but in parallel inhibited their migration, thus delaying the wound healing process [45]. On the contrary, other studies showed that an increased expression of β-catenin was associated with the generation of new epidermis in healing wounds and enhanced motility of HaCaT keratinocytes in vitro [46,47]. Interestingly, β-catenin was shown to have a role in activating the stem cells of the inter-follicular epidermis and the bulge region of hair follicles [48,49]. These stem cell populations contribute significantly to the wound healing process and the restauration of skin original architecture. Namely, the Wnt signaling seems to have a central role in maintaining the hair-inducing activity in skin repair [18]. For example, β-catenin was found to promote stem cell differentiation into follicular keratinocytes, whereas its absence induced cell differentiation into epidermal keratinocytes [18]. It has already been reported that exosomes of ASCs could activate the Wnt/β-catenin pathway in fibroblasts [50], HaCaT cells [29], and in vivo diabetic wounds in mice [51]. However, the effects of ASCs and ASC-CM on the β-catenin expression in the epidermis and the outcomes on tissue regeneration have not been fully investigated yet. Especially, data on humans are missing. In this study, using the ex vivo wound model, we could measure the effects of ASC-CM and ASCs on β-catenin expression and correlate it with the process of re-epithelialization. Our findings suggest that ASCs and ASC-CM improve re-epithelialization and that the Wnt/β-catenin signaling pathway might be involved in this process. Future investigations should focus on the mechanisms of how ASCs and ASC-CM may directly or indirectly influence β-catenin expression in the epidermis of human ex vivo wounds.

### 4.3. Effects on Growth Factors

Neovascularization and re-epithelialization are the major processes in the regenerative phase of wound healing. Growth factors are important players in these processes and their use in wound healing therapies has been extensively investigated [52,53]. For this reason, we measured the levels of selected growth factors in the tissue extracts and culture media of ex vivo wounds. VEGF, PDGF-AA, HGF, and FGF-basic could be measured in samples from all donors, TGF-β1 in only two of six donors, whereas PDGF-BB and EGF could not be detected. The deficiency of PDGF-BB, EGF, and partially of activated TGF-β1 might be due to the fact that the ex vivo skin model is devoid of blood circulation and, thus, of important cell populations like platelets, neutrophils, and macrophages, which are the major source of these factors [38,54]. This might be a drawback of the ex vivo wound model. Nevertheless, comparisons between acute and chronic wounds have found that most of chronic wounds had low levels of TGF- β and EGF [55,56]. Thus, we may consider ex vivo wounds as a model resembling some of the features of poorly perfused chronic wounds. Surprisingly, with respect to control significant increments of VEGF were observed only in tissue extracts and medium of samples treated with DMEM but not in those treated with ASC-CM and ASCs ( Figure 10 and Figure 11). This effect could be due to components of FCS contained in DMEM such as growth factors, antibodies, and hormones, which are necessary for cell survival and proliferation [57,58]. In the ASC samples, which received the stem cells plus DMEM with 10% FCS, this effect was not visible, probably due to consumption of FCS by the ASCs.

It has been reported that ASCs can release several growth factors [6,59]. The presented results showed the capacity of topically applied ASC-CM and ASCs to increase the concentrations of some growth factors in tissue as well as culture media. Nonetheless, different effects were found depending on the treatment: ASCs increased PDGF-AA and HGF and clearly induced higher levels of FGF-basic, whereas application of ASC-CM resulted in significant increments of HGF and a clear increase in PDGF-AA but not of FGF-basic (Figure 10 and Figure 11). This finding may indicate that ASCs seeded on ex vivo wounds were stimulated to produce and release FGF-basic upon contact with the tissue environment and stimulation by tissue mediators such as pro-inflammatory cytokines. For example, it has been shown that exposure of ASCs to TNF-α resulted in increased transcription of proangiogenic factors including FGF-basic [60]. In contrast, ASCs cultured in a non-inflammatory environment were not stimulated to release such factors. Another possible explanation for the different outcomes of the two treatments is the differentiation of ASCs upon contact with the wound tissue [61]. It was reported that ASC-mediated vascularization during tissue regeneration is primarily achieved through the secretion of growth factors in a paracrine fashion, rather than by differentiation into endothelial cells [62]. On the other hand, differentiation of ASCs into endothelial and epithelial cells has been shown in vitro as well in vivo [63]. Detectable co-localization of ASCs and endothelial cell markers in healing wounds of rats has suggested that ASCs might accelerate neovascularization by differentiation into that cell lineage [64]. Another study using GFP-expressing ASCs demonstrated that these stem cells were able to differentiate into epithelial or endothelial cells according to their localization in the wound [65]. To verify the presence of ASCs on the top and in the wound, we stained sections with the stem cells markers CD29 and CD44 (Figure 3). We could observe ASCs mainly on the top of the wound. However, further analyses are necessary to verify the possible differentiation of ASCs in human wounds.

### 4.4. Effects on Inflammatory Mediators

Wound healing is a highly orchestrated process that is coordinated not only by various growth factors but also by immunoregulatory cytokines. Mesenchymal stem cells have been shown to modulate tissue regeneration also by the regulation of the inflammatory process. It has been shown that MSCs exert immunosuppressive effects by releasing immunoregulatory cytokines [66,67]. Similar immunomodulatory properties have been reported also for ASCs [68]. Thus, we also measured the effects of ASCs and ASC-CM on the levels of selected cytokines in the ex vivo wound model. In general, all investigated cytokines were measured in control as well as treated samples, with higher values for the treated groups in comparison to the untreated control. The highest concentrations were found for IL-6, IL-8, and IL-1α, followed by IL-10. Similar results were obtained previously for intact as well as ex vivo wounds after short culture in the trans-well set-up [31,69]. Lower but constantly increasing concentrations were detected for IL-1β, IL-7, and TNF-α. These low levels might indicate that the main source of IL-1β, IL-7, and TNF-α in wounds is given by platelets and infiltrating immune cells [70].

Similarly for the growth factors, different treatments resulted in different cytokine profiles. The most striking result for the DMEM group was a strongly significant increase in IL-10 levels with respect to control at almost all time points (Figure 11). IL-10 has anti-inflammatory and anti-fibrotic effects and, therefore, it plays an essential role in scar-free wound healing [71]. Its increase in samples treated with FCS-supplemented medium might be ascribed to the FCS components. A similar stimulating effect of FCS towards IL-10 release was reported by Silberer et al. using whole blood [72]. The fact that the ASCs group had no significant increase in IL-10 levels with respect to the DMEM group (except for day 9) would indicate that ASCs consume those factors in FCS that stimulate the release of IL-10 by skin cells.

When considering the ASCs group, significant increases of IL-1α, IL-7, and IL-10 were detected (Figure 10 and Figure 11). IL-7 is secreted by stromal cells, keratinocytes, or epithelial cells, and can promote wound healing [73,74,75]. Moreover, it has been reported that IL-7 enhanced the differentiation of ASCs into lymphatic endothelial cells [75]. Therefore, it is likely that IL-7 also plays a positive role in the regenerative response. IL-1α is a pro-inflammatory cytokine, which is constitutively expressed in keratinocytes and therefore immediately released upon skin injury [76]. Notably, a previous study has demonstrated that, in mice, IL-1α can coordinate with IL-7 to activate γδT cells that in turn promote the proliferation of epidermal as well as hair follicle stem cells [77]. In addition, the same study reported that IL-1α can stimulate epidermal stem cell proliferation through interactions with dermal fibroblasts. In our ex vivo wound model, the secretion of IL-1α was increased with respect to the controls in all treated samples, whereas IL-7 was significantly increased in the ASCs and ASC-CM groups but not in the DMEM group. Thus, our results would confirm the ability of ASCs to modulate the immune response towards an anti-inflammatory and regenerative microenvironment. In the ASC-CM group, in addition to IL-1α, IL-7, and IL-10, small but with respect to control, significantly increased amounts of TNF-α and IL-1β were detected (Figure 10 and Figure 11). These findings are in accordance with previous studies that measured cytokines in conditioned medium of ASCs [78,79,80]. IL-1β is also released by skin cells (e.g., fibroblasts, keratinocytes) as well as by infiltrating immune cells (e.g., macrophages) in response to many pathogen-associated molecular patterns (PAMPs) and danger-associated molecular patterns (DAMPs). TNF-α is another pro-inflammatory mediator with a crucial role in wound healing. On the one hand, in chronic wounds an excess of TNF-α was measured, suggesting that prolonged high levels of this mediator have a negative role in the regenerative process [81]. On the other hand, TNF-α can contribute to the recruitment and activation of immune cells in the early phase of wound healing [82] and promotes epidermal stem cell migration and proliferation along with the formation of new hair follicles [83]. Interestingly, it has been shown that TNF-α and LPS can activate bone marrow stromal cells to release prostaglandine-2 that in turn acts on macrophages and induce the release of IL-10 [84]. Thus, such a moderate increase in TNF-α and IL-1 β, along with that of IL-1α, IL-10, and IL-7, would indicate that ASC-CM might be beneficial for the wound healing process in the ex vivo wound model.

In summary, we can state that ASC-CM and ASCs can improve the wound healing process by releasing growth factors and cytokines that modulate tissue repair and regeneration. Interestingly, different growth factors and cytokine patterns were found between the ASCs and the ASC-CM groups (Figure 10 and Figure 11), indicating that the culture conditions and the wound local environment play a crucial role in the activation and secretory behavior of ASCs.

## 5. Conclusions

Wound healing is a complex and multifactorial process. The ex vivo wound model used in this study exhibited some of the characteristics of low-perfusion chronic wounds and allowed us to compare the effects of topical treatments with regard to re-epithelialization and modulation of growth factors and immunoregulatory cytokines. The topical applications of ASCs and ASC-CM were beneficial for the formation of new epithelial cell layers and improved the integrity of the ex vivo skin tissue. Our study demonstrates that the ex vivo wound model can be used as an effective pre-clinical assay platform to investigate the effects of stem cells and their secretome towards the skin cells in chronic wounds. The exact mechanisms by which ASCs and ASC-CM promote immunoregulation and regeneration remain unclear yet. Further investigations to improve the healing potential of ASCs should clarify what influence components of the wound microenvironment have on the ASCs’ secretome. The possibility to control the mediators secreted by stem cells would allow production of ASC-CM or extracellular vesicles with improved immunomodulatory and regenerative properties.

## Figures and Tables

**Figure 1 cells-11-01198-f001:**
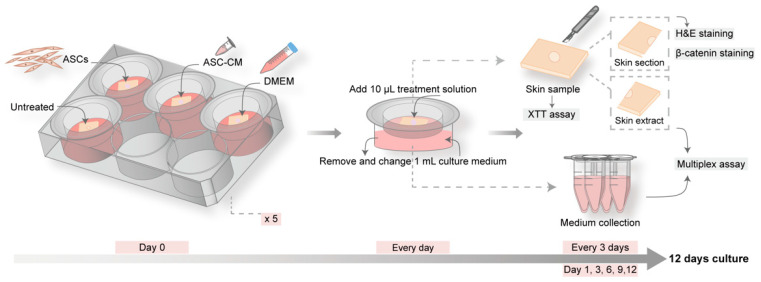
Schematic of the experimental protocol for the ex vivo wound model. Five culture plates were prepared with four skin pieces with one wound in the center. One group served as control and was left untreated, while the other three groups were topically treated with DMEM, conditioned medium of adipose-derived stem cells (ASC-CM), or adipose-derived stem cells (ASCs). Every day, 1 mL of medium was replaced and 10 µL of conditioned medium or DMEM were added to the ASC-CM group or the DMEM and ASCs groups, respectively. At day 1, 3, 6, 9, and 12, one piece of skin sample and 1 mL of culture medium per group were collected. The skin pieces were used for the measurement of skin reductive activity (XTT test). Subsequently, the samples were cut into halves. One half was used for skin sectioning and histology, hematoxylin and eosin (H and E), and β-catenin staining. The other half was processed for protein extraction. Extracts and collected culture medium were used for multiplex analysis of growth factors and cytokines.

**Figure 2 cells-11-01198-f002:**
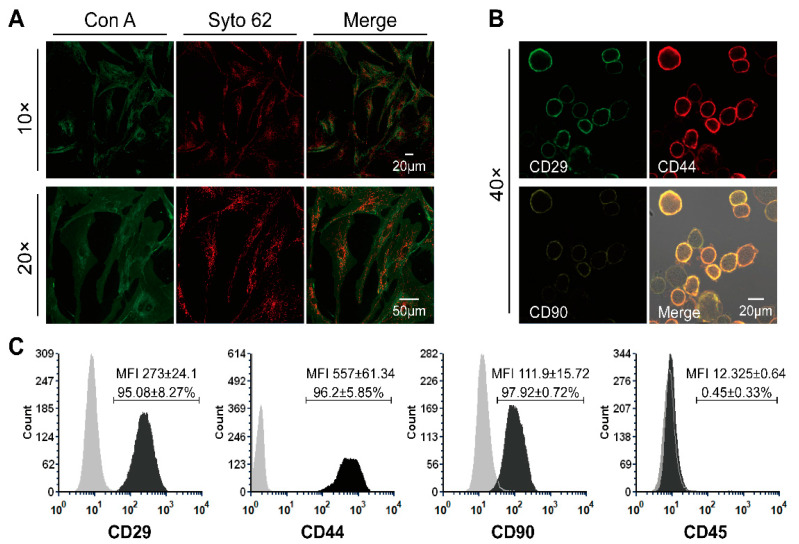
Morphology and immunophenotype of ASCs. (**A**) Representative images of ASCs’ morphology, acquired by the confocal microscope when adherent ASCs reached 80% confluency at passage 4. The cell plasma membrane was stained with concanavalin A (Con A, green), while cell perinuclear membranes were stained in red with Syto62. (**B**) Images of immune-stained ASCs used in the flow cytometry analysis. (**C**) Immunophenotype of ASCs at passage 3 analyzed by flow cytometry (CD29+, CD44+, CD90+, CD45−). Averages ± standard deviations of positive cells and MFI values from three independent experiments are also reported.

**Figure 3 cells-11-01198-f003:**
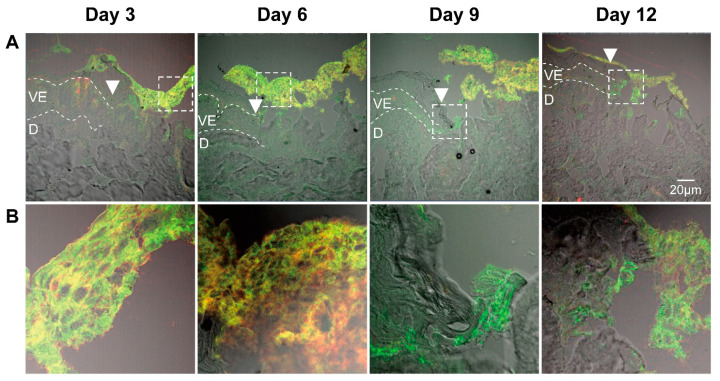
Identification of ASCs in wounds after topical application. Sections of wounds after different incubation days with topically applied ASCs. (**A**) Images of the wound edges taken with 20× magnification. Dotted lines show the viable epidermis and white arrowheads point to the wound edges. VE, viable epidermis; D, dermis. All pictures were taken with the same magnification; scale bar, 20 μm. (**B**) Five time magnification of the boxed areas in (**A**).

**Figure 4 cells-11-01198-f004:**
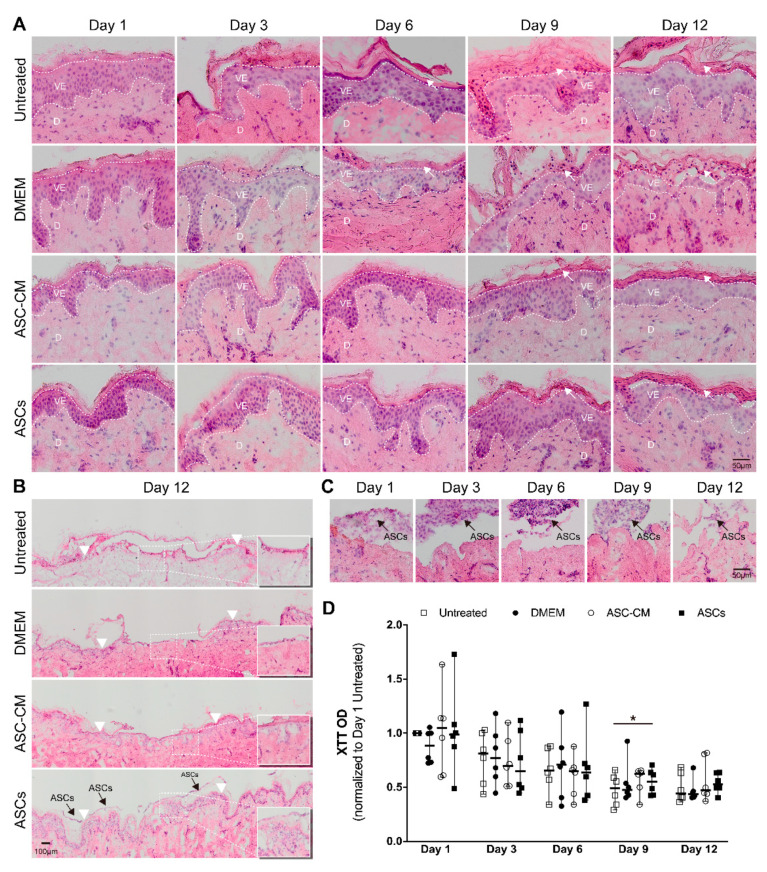
Skin morphology, re-epithelialization, and reductive activity of ex vivo skin samples at different time points during the treatment with DMEM, ASC-CM, and ASCs. (**A**) Representative images of H and E-stained skin sections showing the epidermis close to the wound edge. The white dotted lines show the viable epidermis, whereas the white arrows point to differentiated cells in the upper epidermis. All pictures were taken with the same magnification; scale bar, 50 μm. VE, viable epidermis; D, dermis. (**B**) Representative images showing vertical sections of whole wounds after 12 days of culture. All pictures were taken with the same magnification; scale bar, 100 μm. Insert images are two-fold magnifications of the boxed area. White arrowheads point to the former wound edges. Black arrows point to ASCs. (**C**) Images of ASCs treated wound area at each time point. Black arrow points to ASCs. All pictures were taken with the same magnification; scale bar, 50 μm. (**D**) XTT assay ran with skin biopsies from six independent donors, showing the reductive activity of skin biopsies. Reported values are normalized with respect to the negative control group at day 1. Median and range values are also depicted. The Wilcoxon signed-rank test was used for the statistical analysis. * *p* < 0.05.

**Figure 5 cells-11-01198-f005:**
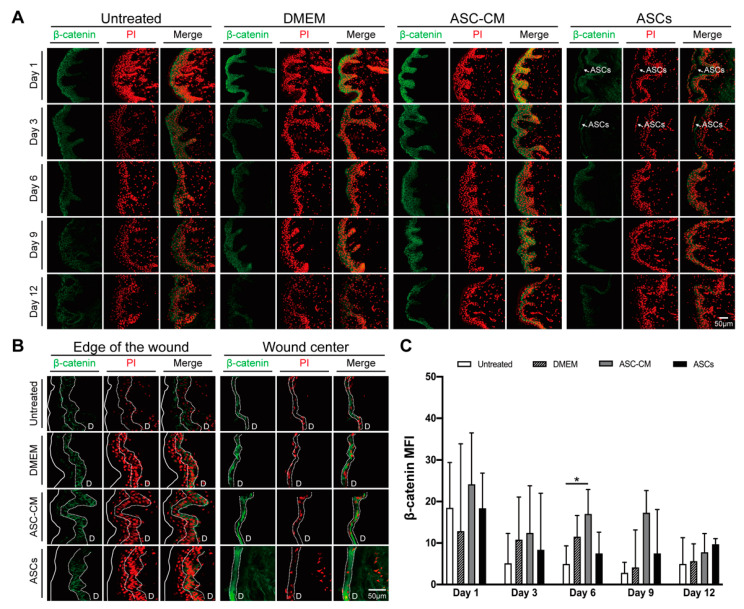
Time-dependent expression of β-catenin in the ex vivo wounds. (**A**) Immunofluorescence staining of β-catenin protein expressed in the epidermis close to the wound edges. Representative images of stained cryosections (β-catenin in green, PI-stained cell nuclei in red, and overlay) for each group and time point. White arrow points to ASCs. All pictures were taken with the same magnification; scale bar, 20 μm. (**B**) Representative immunostained sections of newly formed epithelia at the edge and in the center of wounds at day 12. The top of the wound is indicated with a continuous line while the newly formed epithelia is indicated by dashed lines. All pictures were taken with the same magnification; scale bar, 20 μm. D, dermis (**C**) Analysis of the fluorescence intensity of at least 15 random fields per time point from three (control) or six donors (samples). The mean fluorescence intensity (MFI) was measured using the ImageJ software. Data are shown as median with interquartile range (IQR). The statistical analysis was performed using the Mann–Whitney U test. * *p* < 0.05.

**Figure 6 cells-11-01198-f006:**
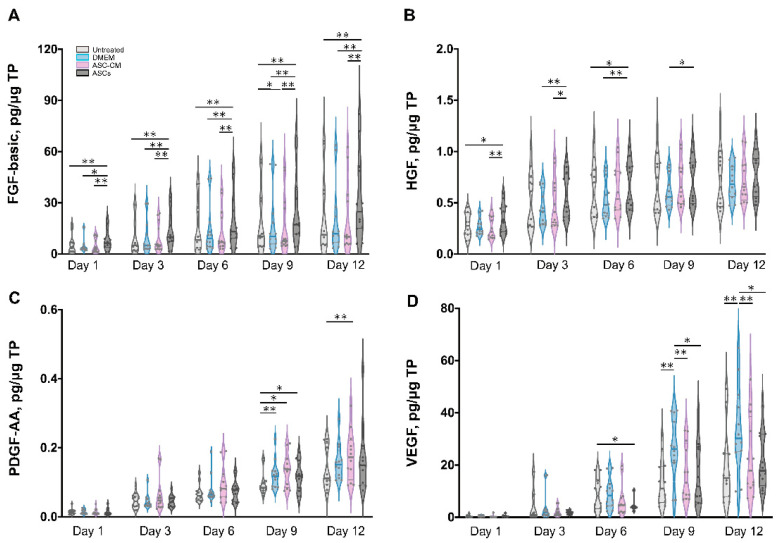
Quantification of growth factors in skin extracts. The levels of FGF-basic (**A**), HGF (**B**), PDGF-AA (**C**), and VEGF (**D**) were measured in samples collected from six donors at different time points. Each sample was measured in duplicate. The values for the different groups were normalized to the respective sample total protein and reported as the cumulative amount of analyte versus time. The thin gray lines show the 75th percentile (upper) and 25th percentile (lower); the thicker bars within the violin plots indicates the median. Statistical analysis was done using the Wilcoxon signed-rank test. * *p* < 0.05, ** *p* < 0.01.

**Figure 7 cells-11-01198-f007:**
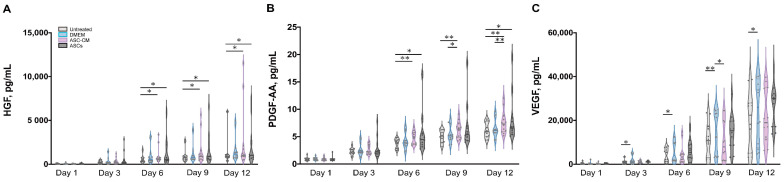
Quantification of growth factors in culture medium. The levels of HGF (**A**), PDGF-AA (**B**), and VEGF (**C**) were measured in medium collected from six donors at different time points. Each sample was measured in duplicate. The values for the different groups are reported as cumulative amount of analyte versus time. The thin gray lines show the 75th percentile (upper) and 25th percentile (lower); the thicker bars within the violin plots indicates the median. Statistical analysis was done using the Wilcoxon signed-rank test. * *p* < 0.05, ** *p* < 0.01.

**Figure 8 cells-11-01198-f008:**
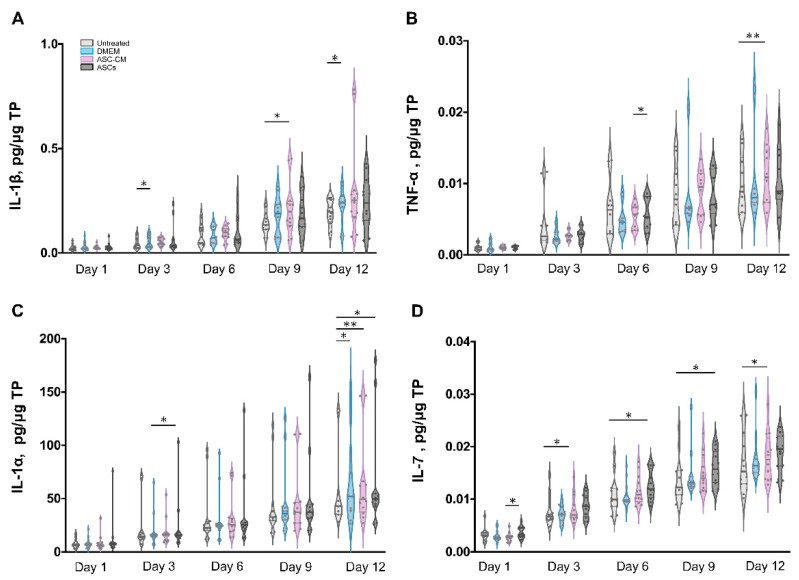
Quantification of cytokines in skin extracts. The levels of IL-1β (**A**), TNF-α (**B**), IL-1α (**C**), and IL-7 (**D**) were measured in samples collected from six donors at different time points. Each sample was measured in duplicate. The values for the different groups were normalized to the respective sample total protein and reported as cumulative amount of analyte versus time. The thin gray lines show the 75th percentile (upper) and 25th percentile (lower); the thicker bars within the violin plots indicates the median Statistical analysis was done using the Wilcoxon signed-rank test. * *p* < 0.05, ** *p* < 0.01.

**Figure 9 cells-11-01198-f009:**
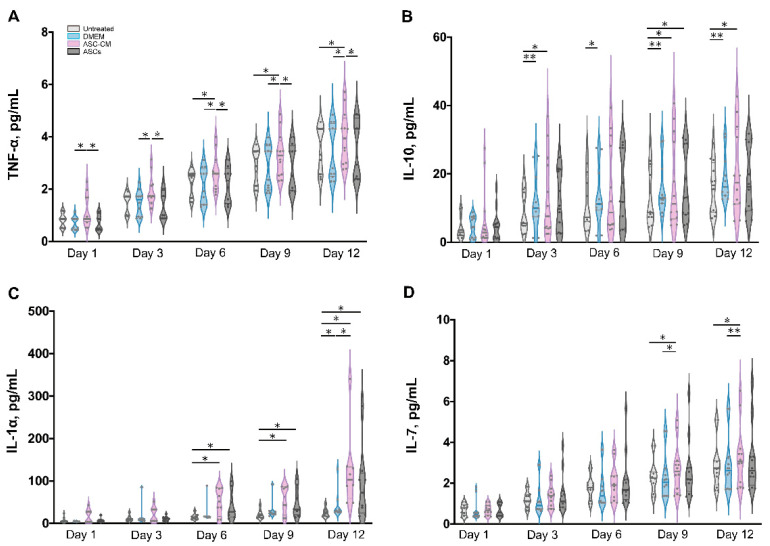
Quantification of cytokines in culture medium. The levels of TNF-α (**A**), IL-10 (**B**), IL-1α (**C**), and IL-7 (**D**) were measured in medium collected from six donors at different time points. Each sample was measured in duplicate. The values for the different groups are reported as the cumulative amount of analyte versus time. The thin gray lines show the 75th percentile (upper) and 25th percentile (lower); the thicker bars within the violin plots indicates the median. Statistical analysis was done using the Wilcoxon signed-rank test. * *p* < 0.05, ** *p* < 0.01. IL-1 α was detected in only four of six donors.

**Figure 10 cells-11-01198-f010:**
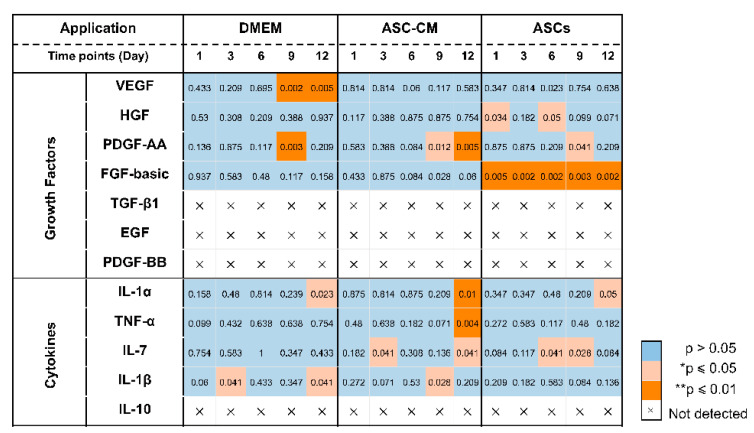
Visualization of statistically significant differences between treated samples and controls in wound extracts.

**Figure 11 cells-11-01198-f011:**
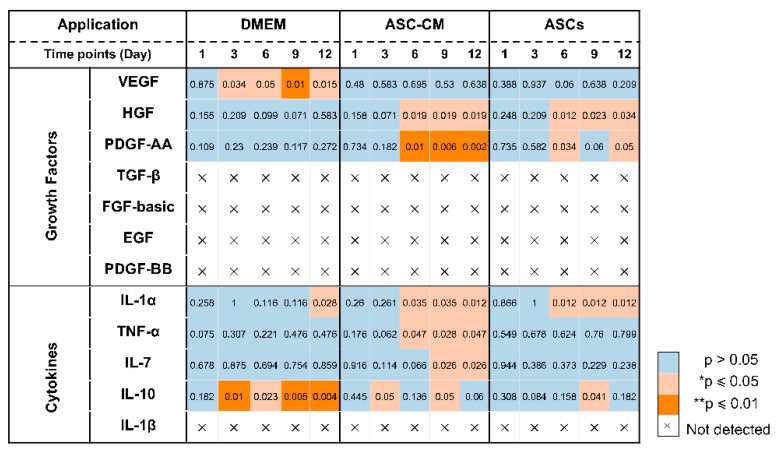
Visualization of statistically significant differences between treated samples and controls in wound culture media.

## Data Availability

The data used to support the findings of this study are included within the article.

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
