# Peer review of "Effects of Adipose-Derived Stem Cells and Their Conditioned Medium in a Human Ex Vivo Wound Model"

_cells, 2022, doi:10.3390/cells11071198_

Round 1
Reviewer 1 Report
The present manuscript, "Effects of Adipose-derived Stem Cells and Their Conditioned Medium in a Human ex vivo Wound Model" is based on the novel concept. The manuscript is well constructed and written. However, a minor point needs to be addressed before acceptance.
- In fig. 2, the authors need to provide the quantitative data on the CD29+, CD44+, CD90+, CD45- cells from immune fluorescence and flow cytometry analysis (MFI values).
Reviewer 2 Report
The use of language is efficient. The literature review showed that there are several research articles already published with similar headlines, but the cell lines used are different. The idea is not quite unique.
What is your main difference between literatures and the reason why this manuscript is unique to published?
The results are quite promising. Introduction part is detailed and gives the reader all the background information needed.the figures are clearly explained and the datas are compared good. Figures are labelled with arrows and lines for focusing the reader to the needed parts.
1. Line 44, choric wounds, typing mistake “chronic”.
2. Line 132, FITC can be written with the longer version.
3. Until line 220-221 H&E always mentioned with only abbreviation not as the long
version. Long version must be added at the first the the abbreviation is used.
4. Figure 2 the images can be enlarged for better observations.
5. Figure 5 and 6 can be plotted by using violin plot for better visualization.
6. Is there a specific reason why the authors choose the articles which don’t have recent
dates?

Reviewer 3 Report
In this article, Guo X et al investigate the effects of adipose stem cells in a human ex vivo wound model. Using this type of in vitro model system is important for the development of better tools to treat humans, compared to the use of animal models that have some limitations.
Unfortunately, although the authors have made a big effort in the establishment of this model, there are several problems that should be addressed.
The authors use adipose stem cells for the treatment, but no sign of engraftment or analysis of the localization of these cells is provided. Using a GFP to trace them would have been useful.
The last figures with the QPCRs should all be edited and combined in a better way, including the heatmaps together with the data they refer to.
Finally, the authors should provide some functional outcome, that is not only based on the expression of several genes by QPCR. It will be interesting to see if the cells remain with any of the treaments, or if anything else is changing within the histology at the last timepoint.
Reviewer 4 Report
The author's have done good job on the study. They have supported the hypothesis with data. Here are few suggestions:
1) Author's never introduce why beta-catenin is important and why they are studying it? It suddenly appears in methods and results.
2) Please include Skin exoplant in culture pictures for all the treatment groups at each time point to have visual representation.
3) Line 326, 'ROW is written as 'RAW'. Please check the entire manuscript for such minor mistakes.
4) Please include scale bar in all the images.
5) Use arrows or dotted lines to show where the wound was and how it has healed.
6) Labeling of skin H&E slides with various layers of skin will add more information for the reader.
7) Please clarify the total number of patients used. 20 skin samples were used for 4 different treatments groups whereas in statistical methods, authors point toward using 15 donors only? which one is correct?
8) MSCs seem to work through canonical wnt signaling pathway for almost everything, it will be great if authors could explore more in this pathway and show actual relevance to their study.
Round 2
Reviewer 3 Report
The authors have answer my comments appropiately.
However, I consider Figure Supplementary 1 should be moved to the Main Figures since it has very important information. They can either add it as an extra Figure or merge it with another one.